# Birth preparedness and complication readiness knowledge, practices and its associated factors among recently delivered women: A cross-sectional study in Bharatpur, Chitwan, Nepal

Asmita Ghimire[1,¤a]*, Bipan Bahadur Tiwari[1,¤b], Eak Narayan Poudel[1‡], Mamta Chhetri[1‡], Devraj Regmi[1‡], Laxman Poudel[2‡]

1 School of Public Health, Chitwan Medical College, Tribhuwan University, Chitwan, Bagmati, Nepal,
2 Health Section, Sitganga Municipality, Arghakhachi, Lumbini, Nepal

☯ These authors contributed equally to this work.
‡ ENP, MC, DR and LP also contributed equally to this work.
¤a Current address: Health Section, Chhatradev Rural Municipality, Arghakhachi, Lumbini, Nepal
¤b Current address: Nepal Development Society, Bharatpur, Chitwan, Bagmati, Nepal
* ghimireasmita570@gmail.com

## Abstract

### Background

Birth preparedness and complication readiness (BPCR) is a comprehensive strategy, aimed at ensuring that expectant mothers and their families are for normal delivery and preparing for potential complications that may arise during pregnancy, labor, delivery, and the postpartum period without any delays. BPCR interventions are widely promoted by government and international agencies to reduce maternal and neonatal health risk in developing countries like Nepal. Studying BPCR also helps identify gaps in knowledge, access, and practices, guiding interventions to strengthen health system and community awareness, especially in low-resource settings.

### Objective

This study was conducted to assess birth preparedness and complication readiness knowledge, practices and its associated factors among recently delivered women in Bharatpur city, Chitwan, Nepal.

### Methods

A community based cross-sectional study was conducted in 2022, on a sample of 220 recently delivered women. Data were collected using pre-tested structured interview questionnaire. The collected data were analyzed by IBM SPSS 20 version software. Variables with p-value ≤0.05 on the bivariate analysis were included in multivariate analysis. Adjusted odds ratios (AOR) with the respective 95% Confidence

**Data availability statement:** Data Availability Statement has been added to the manuscript with its URL, repository name. The raw data has been made available through given URL below: https://figshare.com/s/3199a871d0b-deae608e3 DOI: [https://doi.org/10.6084/m9.figshare.26213132].

**Funding:** Authors name who received awards: Asmita Ghimire Funded by: CMC Research grant Committee URL :https://www.cmc.edu.np/cmc-irc The study was funded by Chitwan Medical College Research Grant Committee. However, the funder had no role in study design, data collection and analysis, decision to publish, or preparation of the manuscript.

**Competing interests:** The authors have declared that no competing interests exist.

Interval (CI) and a p-value <0.05 was used to set statistically significant variables in the multivariable analysis.

## Results

Among 220 recently delivered women, majority (91.4%) of the women identified the place of delivery and saved money for childbirth (97.7%). Similarly, most of them arranged transportation (87.3%), identified a companion (85.9%), and arranged necessary materials (90%) for childbirth. Considerable (52.3%) women identified skilled birth attendants. Preparedness for blood donors (36.4%) found to be low as compared to other components. Overall, 46.8% of recently delivered women were well prepared. Family type, knowledge on components of BPCR, obstetric signs and symptoms and ANC visits were found to be statistically significant (p-value<0.05) with birth preparedness practice. Associated characteristics were further subjected to multivariate logistic regression where knowledge on components of BPCR (AOR = 5.34,95%CI: (2.728–10.478) was found significantly associated with BPCR practice.

## Conclusion

The overall 46.8% of women who prepared for birth and its complication readiness was found to be higher as compared to other reports. Encouraging women to utilize antenatal care, and such as sensitization of pregnant women, during ANC visit by the health workers, regarding components of BPCR, danger sign and symptoms during pregnancy, may enhance BPCR.

## Introduction

Reducing maternal mortality is a key priority on the international agenda. The new global target is to lower the maternal mortality ratio (MMR) to fewer than 70 maternal deaths per 100,000 live births, with each country aiming to reduce their MMR by at least two-thirds from the 2010 baseline. Additionally, by 2030, no country should have an MMR exceeding 140 deaths per 100,000 live births [1–3]. A safe motherhood strategy namely "Birth preparedness and complication readiness" aims at reducing maternal mortality and morbidity [4], was incorporated in the World Health Organization (WHO) antenatal care package in 2005 [5].

Each day, for thousands of women and their families, the event of childbirth becomes a reason of unnecessary suffering due to acute obstetric complications and maternal deaths [6]. United Nation Children Fund Organization (UNICEF) and World Health Organization (WHO) reported that (585,000) women die annually following complications related to pregnancy and childbirth and 99% of these are in developing countries with about 70% of these mortalities occurring in sub-Saharan Africa [7]. The primary reasons for such mortality were often linked to underutilization of available maternal health services during pregnancy and inadequate preparation for possible complications related to childbirth [8]. Given the unpredictable nature of childbirth and

the associated risk, providing prompt and adequate medical care to women experiencing obstetric complication is vital for reducing these risks. To ensure timely access to skilled maternal and neonatal care during pregnancy, labor, and delivery, it is essential to identify and address any delays. One effective approach to achieve this is through Birth Preparedness and Complication Readiness (BPCR), a core element of globally endorsed safe motherhood initiatives [9,10].

Birth Preparedness and Complication Readiness (BP/CR) is a proactive approach that involves planning for a safe, normal birth while being ready to act swiftly in case of emergencies. This comprehensive strategy empowers women, their families, and the community to improve birth planning and respond effectively to emergencies, reducing delays and ensuring better outcomes [11]. It is estimated that exposure to BPCR interventions results in 18% reduction in neonatal mortality risk and a 28% reduction of in maternal mortality risk [12]. So as to reduce the risks associated with pregnancy and childbirth and address factors associated with mortality and morbidity, BPCR strategies have been adopted in Nepal [13] which was rolled out in all 75 districts in 2008–2009 to improve timely access to delivery care services [14].

Birth preparedness motivates people to take proper care during pregnancy and ensure a skilled care provider at every birth. Complication readiness raises awareness of danger signs among women, families and community and prepares them to respond in a proper manner during emergencies [15]. It promotes health care seeking behavior and utilization of appropriate heath care facilities and skilled personnel for delivery and hence reduce maternal death [16]. In the literature, a host factors were reported as determinants of BPCR such as; knowledge of danger signs of pregnancy, educational level, employment status, adequacy of ANC visits, maternal age, place of residence, poverty and birth order [17].

The goal of reducing maternal mortality and morbidity has fallen woefully behind [18]. According to UN inter-agency, the global maternal mortality ratio declined by 38 per cent – from 342 deaths to 211 deaths per 100,000 live births, in 2000–2017 which is less than half to achieve the Sustainable Development global goal [19]. Every day in 2017, approximately 810 women died from preventable causes related to pregnancy and childbirth [20].

Among South Asian regions, India had the highest estimated numbers of maternal deaths, accounting for 12% of global maternal deaths. The proportion of BPCR practices varied from 47% in India, less than a quarter (24.5%) of women were considered well prepared for birth in Bangladesh, 59% in Sri-Lanka, and 32% had not practiced any components of BPCR in Nepal [12,21–23].

Studies from Nepal, Burkina Faso, India, and Nigeria demonstrate that promoting BPCR has increased women's awareness of danger signs and positively influenced their health-seeking behavior during obstetric emergencies [24–27]. Hence birth preparedness and complication readiness plan is recommended with the notion of pregnancy is risky [28].

In spite of government and non- government organization focused on to reduce MMR and organized many pro-grammes [29], Nepal is also one of the highest MMR countries in the world with 239 per 100,000 live births in 2016 [4]. Approximately (12%) of deaths among women of reproductive age were classified as maternal deaths in 2016, which most could have been prevented through proper practices of BPCR [30]. Only 15% had arranged for transport, (62%) had saved money, (2%) had identified potential donor, (2%) identify and contact health facilities and health workers while16% had not made any preparations at all [31].

Despite the fact that birth preparedness and complication readiness is essential for further improvement of mater-nal and child health little is known about the current magnitude and influencing factor in Nepal. To reduce MMR women should prepared for different components of BPCR under Safe motherhood program. This study will be help to assess the status of BPCR practice, fill gaps and might be helpful for future researchers and local level authority for planning and implementing BPCR program for women of the community.

## Materials and methods

### Study area

Bharatpur is a city in the central-southern part of Nepal. Located in Chitwan district, Bharatpur is the district headquarter of the Chitwan district, as well as a separate Metropolitan authority, and is the fifth largest city of Nepal. There are total of

 

29 wards and the study was conducted in three wards, 14, 15 and 16 of Bharatpur Metropolitan City. The reason behind the selection of these wards were: the only Primary Health Care of Bharatpur Metropolitan City located in ward no 14 and the patient flow also high as compared to others. These wards span an area of 12.07, 20.5 and 17.06 square kilometers and comprises, 2094, 1845 and 1894 households respectively, accommodating a total of 33,988 individuals.

## Study design

A community based cross-sectional study was conducted using quantitative methods to assess the knowledge and practices related BPCR from September 25 to December 29, 2022. The study was conducted among the recently delivered women who were visiting the three governmental health institutions for immunization of their child within selected wards of Bharatpur, Chitwan during the study period. Women who volunteered to participate and responded to the questionnaire were included whereas, exclusion from the study encompassed women who seek vaccination for MR 2$^{nd}$ dose and typhoid (these vaccine are administered at 15 months of age as per out National Immunization Schedule) with mental illness, and those unwilling to provide written consent were excluded. This design was chosen because it involves a large participant pool, provides comprehensive details, and generates precise and targeted responses.

## Sample size and sampling method

The study's sample size was determined by assuming of 95% confidence interval, with 6% margin of error considering the similar study conducted in similar district with prevalence of poor BPCR practice (29%) [32]. A 6% margin of error was chosen as a balanced compromise between the standard 5% (which requires a larger sample size and 7% (which offers less precision). Given the expected prevalence of 29%, 6% provides sufficient accuracy for the study's objectives while remaining feasible in terms of time, cost, and resources.

This gave a computed value and the total sample size became 220,

$$n_0 = \frac{z^2 pq}{d^2}$$

$$= \frac{1.96^2 * 0.29 * 0.71}{0.06^2}$$

$$= 220$$

Non- Probability convenience sampling technique was used to select study subjects. Firstly, ward no. 14, 15 and 16 were selected purposively as a study site and the total number of governmental health institutions within wards where immunization conducted: health post, Primary health care, out-reach clinic with their respective immunization date were identified. From the above information, three health institution were selected from each ward namely: Shivanagar Primary health care (only PHC in metropolitan city), Fulbari health post and Mangalpur health post along with their out-reach clinic, and required sample from each health institution was selected conveniently. These three sites were chosen due to higher patient flow, representative diversity in the population characteristics of peri-urban areas, operational feasibility. In this study, covenience sampling involved selecting individuals who were readily accessible, available and willing to participate at the time of data collevtion. The data was collected through face-to-face interview after taking written informed consent from the participants.

## Measurement

The pretested structured questionnaire was partially adapted from standard tools developed by the World Health Organization (WHO) and JHPIEGO's Maternal and Neonatal Health Program, with additional items developed by the authors based on the study objectives and local context. To achieve content validity, items were drawn from the previously validated survey and developed based on an extensive literature search with frequent review to enhance their quality. The questionnaire underwent a pre-testing phase in a different ward of Bharatpur Metropolitan City in the Chitwan district, excluding ward no 14,15 and 16. Necessary adjustments were made accordingly while facilitating accessibility by translating the questionnaire back and forth between Nepali and English languages.

Using a pre tested questionnaire the following information were collected. Outcome variables includes Level of Birth Preparedness Practices as well prepared and less prepared. Socio demographic characteristics including age, ethnicity, religion, education, occupation and average monthly income. The questionnaire included knowledge related questions as knowledge on BPCR components, danger signs and symptoms during pregnancy, labour and postpartum, ANC visits.

The women were asked about obstetric factors, institutional factors and role of family and neighbors.

## Operational definitions

**Level of BPCR Practice:** A woman was considered "well prepared" for birth and its complications when she practiced first four critical components among seven basic components **(i)** identified transportation for delivery, ii) saved money, iii) identified health facility, iv) identified blood donor v) identified skilled health provider, vi) identifying labour and vii) birth companion, preparing necessary materials for mother and baby) [25].

**BPCR components knowledge:** A woman was considered adequate knowledge on BPCR components when she spontaneously mentioned at least four components.

**Knowledge on key danger signs during pregnancy:** A woman was considered having adequate knowledge, if she spontaneously mentions at least three danger signs: vaginal bleeding, swollen hands/face, blurred vision, severe abdominal pain, reduced/ no fetal movement, smelling discharge during pregnancy.

**Knowledge on key danger signs during labor:** A woman was considered having adequate knowledge, if she spontaneously mentions at least three danger signs: severe vaginal bleeding, severe headache, prolonged labor (12huors), convulsion, and retained placenta during labor/childbirth.

**Knowledge on key danger signs during postpartum:** A woman was considered having adequate knowledge, if she spontaneously mentions at least three danger signs: severe vaginal bleeding, foul-smelling vaginal discharge, and high fever, unless considered as inadequate knowledge.

## Data collection process

The data was collected through face-to-face interview during the immunization date in selected wards institution along with their out-reach clinic which was held from 9–13 of every month, before immunizing the child of respective mother only after getting written informed consent.

## Data analysis

The collected data were coded, entered, cleaned and analyzed by IBM SPSS 20 version. Firstly, descriptive statistics, such as frequencies, percentages, and medians were computed to provide an overview of data. Practice level was divided into well prepared and less prepared where, those mothers who followed four critical components among seven were considered "well prepared" and the remaining were considered "less prepared". Components of BPCR are**)** identified transportation for delivery, ii) saved money, iii) identified health facility, iv) identified blood donor v) identified skilled health provider, vi) identifying labour and vii) birth companion, preparing necessary materials for mother and baby). For

knowledge variables, knowledge was assessed for each components individually, based on spontaneous mention of correct responses as the criterion mentioned in the operational definition. Additionally, inferential statistics were employed, specifically the Pearson chi-square test at a significance level of 5%, to explore potential associations between level of practice and selected explanatory variables. A multi-collinearity diagnostic test was conducted to check for collinearity among the independent variables using VIF with the threshold of 10 and there was no multicollinearity detected between the variables.

## Data quality control

To maintain the validity of the measurement, questions were drawn from the previously validated survey and developed based on an extensive literature search with frequent review to enhance their quality. The questionnaire underwent a pre-testing phase in other ward of Bharatpur Metropolitan City in the Chitwan district, excluding ward no 14,15 and 16. Necessary adjustments were made accordingly while facilitating accessibility by translating the questionnaire back and forth between local language (Nepali) and English.

## Ethical considerations

Ethical approval was obtained from Thesis Committee of the School of Public Health, Chitwan Medical College and Ethical Review Committee of Chitwan Medical College and from Chitwan Medical College Institutional Review Committee (IRC). (Ref: CMC-IRC/079/080-016). Letter of support was obtained from the health section of Bharatpur Metropolitan Office and respective health institutions before undertaking the study and written informed consent was obtained from the respondents before the interview. For Privacy and confidentiality, all interviews were conducted in private and all cautions were taken to ensure confidentiality. Participants were informed of their rights to deny participation in the study or to withdraw from the study at any stage of data collection. Respondents were provided with the necessary information on importance of birth preparedness and complication readiness to avoid delays in seeking and receiving health care services.

## Results

### Socio-demographic and obstetric characteristics

A total of 220 recently delivered women were interviewed (Table 1). The age range of respondents varied from 17 to 48. By ethnicity 101(45.9%) were Brahmin/Chhetri and 4(1.8%) were Thakuri/Sanyasi. The major predominant religion was Hindu 181(82.3%) with Christianity being 7(3.2%). Occupationally 169(76.8%) were housewives followed by services 10(4.5%). Regarding educational background, 56.6% had secondary education and 12.7% had bachelors and above. The majority of respondents family income 80(36.4%) were between 30–50 thousand rupees.

### Obstetric related variables

Regarding birth order, 48.2% of the women had already given birth for one child while 1(0.5%) had four children. About 215(97.7%) of respondents attended antenatal care in their recent pregnancy. Among those women who attended ANC more than four times were 104(47.3%) and 111(50.5%) less than four visits (Table 2).

Knowledge regarding the components of Birth Preparedness and Complication Readiness (BPCR), 59.1% of women had adequate knowledge (Fig 1). Bar graph depicting the distribution of women's knowledge regarding danger signs associated with pregnancy, childbirth, and the postpartum period. The blue bars represent the percentage of women who demonstrated adequate knowledge by identifying at least three key danger signs: 50.9% for pregnancy, 20.5% for childbirth, and 27.7% for the postpartum period. Conversely, the orange bars indicate the percentage of women with inadequate knowledge, having mentioned fewer than three danger signs: 49.1% for pregnancy, 79.5% for childbirth, and 72.3% for the postpartum period. When asked about knowledge on ANC checkups, 90.5% of women had knowledge about it while 9.5% did not.

**Table 1. Socio-demographic characteristics of recently delivered women in Bharatpur, Chitwan, Nepal (n=220).**

| Characteristics | Frequency | Percent (%) |
|---|---|---|
| **Age** | | |
| 17-25 | 109 | 49.5 |
| >25 | 111 | 50.5 |
| Median (IQR) = 26 (23–29) | | |
| **Family Type** | | |
| Nuclear | 116 | 52.7 |
| Joint/Extended | 104 | 47.3 |
| **Ethnicity** | | |
| Brahmin/Chhetri | 101 | 45.9 |
| Janajati | 66 | 30 |
| Dalit | 42 | 19.1 |
| Madhesi | 7 | 3.2 |
| Thakuri/Sanyasi | 4 | 1.8 |
| **Religion** | | |
| Hindu | 181 | 82.3 |
| Buddhism | 32 | 15.5 |
| Christianity | 7 | 3.2 |
| **Educational level (n=215)** | | |
| General literate | 6 | 2.7 |
| Basic | 56 | 25.9 |
| Secondary | 125 | 56.6 |
| Bachelors and above | 28 | 12.7 |
| **Mother's occupation** | | |
| Housewife | 169 | 76.8 |
| Agriculture | 20 | 9.1 |
| Business | 17 | 7.7 |
| Services | 10 | 4.5 |
| Labor | 2 | 0.9 |
| Others* | 2 | 0.9 |
| **Monthly income level (In Rupees)** | | |
| Less than 10,000 | 12 | 5.5 |
| 10,000-30,000 | 88 | 40 |
| 30,000-50,000 | 80 | 36.4 |
| More than 50,000 | 40 | 18.2 |

Table 3 outlines the most commonly mentioned items of danger signs during pregnancy which were vaginal bleeding 131(65.2%), severe headache 80(39.8%) and swollen hand and face 69(34.3%). During labor and delivery, the majority of listed was severe bleeding 155(86.1), severe headache 68(37.8%) and loss of consciousness 61(33.9%). Similarly, severe bleeding 171(89.1%) was the most commonly mentioned items. The reported 59.1% as having adequate knowledge does not refer to a composite knowledge score. In this study, each knowledge component was assessed separately based on spontaneous mention of relevant information, and no total or cumulative knowledge score was calculated.

## Institutional related factors

Table 4 describes the distance to health and delivery facilities, as well as the availability of 24- hour delivery services with median distance to health facilities being 15 minutes, minimum of 2 minutes and a maximum 35 minutes. The majority of

**Table 2. Obstetric related factors among recently delivered women in Bharatpur, Chitwan, Nepal (n = 220).**

| Birth order | | |
|---|---|---|
| One | 106 | 48.2 |
| Two | 94 | 42.7 |
| Three | 19 | 8.6 |
| Four | 1 | 0.5 |
| **Place of delivery** | | |
| Institutional | 217 | 98.6 |
| Home | 3 | 1.4 |
| **ANC visit** | | |
| Yes | 215 | 97.7 |
| No | 5 | 2.3 |
| **Number of ANC (n = 215)** | | |
| Less than four times | 111 | 50.5 |
| Greater than or equal to four | 104 | 47.3 |
| **Complication experience** | | |
| Yes | 101 | 45.9 |
| No | 119 | 54.1 |
| Complications (101) | | |
| Smelling discharge | 22 | 22.2 |
| Convulsions | 7 | 7.1 |
| Swollen hands and face | 29 | 29.3 |
| Labor lasting more than 12 hours | 7 | 7.1 |
| Severe headache | 13 | 13.1 |
| Vaginal bleeding | 22 | 22.2 |
| Blurring of vision | 2 | 2 |
| Abdominal pain | 38 | 43.3 |
| Others** | 17 | 17.2 |

Other*student, other** vomiting, allergy, diabetes, meconium aspiration syndrome, High blood pressure.

participants reported having access to a health facility within 20 minutes. However, over 80% of participants reported a distance of more than 30 minutes to delivery facilities. Additionally, 61.8% of participants reported that 24-hour delivery services were not available. For those whose nearest health facility did not have delivery/24 hours services, 82.3% of respondents reported that their nearest delivery facility was more than 30 minutes away.

### Family and neighbors support related information

Table 5 presents the support received by the women during delivery from their family and neighbors. About 19.5% of women who made the decision alone,26.8% made with their husbands while 5.9% responds that family members alone made decision for delivery. When asked whether they received assistance from community people, 38.6% of the respondents that they did, while 61.4% did not. For those who did receive assistance, the most common types of assistance provided were arranging transportation (60.2%), arranging money (32.5%), arranged blood and 24.1% reported receiving other types of assistance.

### Association between level of BPCR Practice and some explanatory variables

In this study, the chi-square test was employed to analyze the relationship between the independent variables and the level of preparedness (Fig 2). The results indicated that significant differences in the level of preparedness were observed

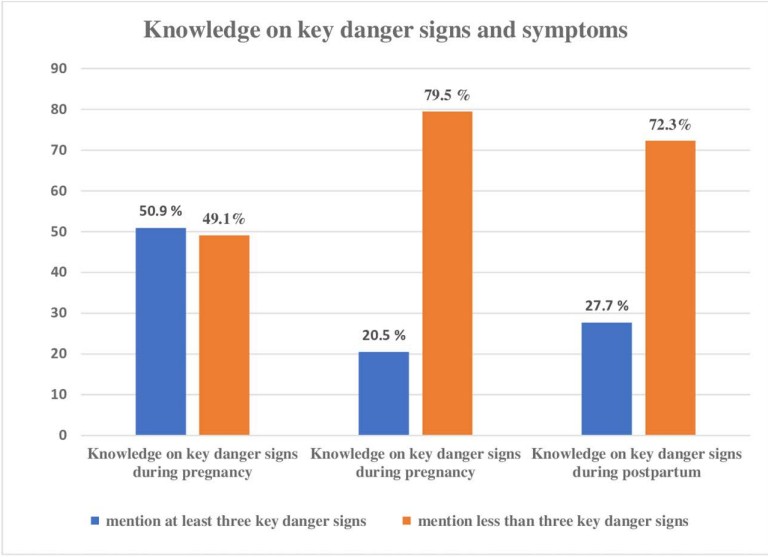

**Fig 1. Participants knowledge about key danger signs.**

**Table 3. Knowledge of respondents on danger signs during pregnancy, labour and post- partum period in Bharatpur, Chitwan, Nepal (n = 220).**

| Danger sings during pregnancy | Frequency | Percentage (%) |
|---|---|---|
| Vaginal bleeding | 131 | 65.2 |
| Severe headache | 80 | 39.8 |
| Swollen hand/face | 69 | 34.3 |
| Blurring of vision | 36 | 17.9 |
| Severe lower abdominal pain | 134 | 66.7 |
| Smelling discharge | 86 | 42.8 |
| Convulsion | 15 | 7.5 |
| Reduced/no fetal moment | 20 | 10 |
| **Danger signs during labor/childbirth** | | |
| Severe bleeding | 155 | 86.1 |
| Severe headache | 68 | 37.8 |
| Loss of consciousness | 61 | 33.9 |
| Labor lasting >12 hours | 56 | 31.1 |
| Placenta not delivered 30 min after baby birth | 32 | 17.8 |
| **Danger signs during Postpartum** | | |
| Severe bleeding | 171 | 89.1 |
| Severe headache | 72 | 37.5 |
| Convulsion | 45 | 23.4 |
| High fever | 92 | 47.9 |
| Foul vaginal discharge | 24 | 12.5 |

**Table 4. Institutional related information among recently delivered women in Bharatpur, Chitwan, Nepal (n=220).**

| Variables | Frequency | Percentage (%) |
|---|---|---|
| **Distance to health facilities** | | |
| Less than 20 min | 168 | 76.4 |
| More than 20 min | 52 | 23.6 |
| *Median(IQR)=15(10–20) MIN, MAX=2, 35* | | |
| **Availability of delivery/24hrs facility** | | |
| Yes | 84 | 38.2 |
| No | 136 | 61.8 |
| **Distance to delivery facilities** | | |
| Less than 30 min | 39 | 17.7 |
| More than 30 min | 181 | 82.3 |

**Table 5. Support by family and neighbors related information among recently delivered women in Bharatpur, Chitwan, Nepal (n=220).**

| Variables | Frequency | Percentage (%) |
|---|---|---|
| **Decision maker for delivery** | | |
| Women alone | 43 | 19.5 |
| Women and husband | 59 | 26.8 |
| Jointly | 105 | 47.7 |
| Family members alone | 13 | 5.9 |
| **Assistant by community people** | | |
| Yes | 85 | 38.6 |
| No | 135 | 61.4 |
| **If yes, what are those things (n=85)** | | |
| Arrange transportation | 50 | 60.2 |
| Arrange money | 27 | 32.5 |
| Arrange blood | 3 | 3.6 |
| Others* | 20 | 24.1 |

*Denotes multiple response: food, cloths.

concerning family type (P=0.025), knowledge on components of BPCR (P=0.001), knowledge on signs and symptoms during pregnancy (P=0.002), knowledge on signs and symptoms during labor/childbirth (P-value=0.008), knowledge on signs and symptoms during postpartum (P=0.05) and knowledge on ANC visit as protocol (P=0.02) (Fig 3, Table 6). However, no significant differences were found in level of preparedness based on age, occupational status, religion, birth order.

**Test of multicollinearity between independent variables and Birth preparedness and complication readiness (BPCR).** In test of multicollinearity, those independent variables which where statistically significant and associated with birth preparedness and complication readiness practices were further used to test the multicollinearity among those independent variables.

**Test of multicollinearity between independent variables and BPCR practice among recently delivered woman in Chitwan, Nepal (n=220).** The data includes variables like family type, BPCR knowledge, danger sign awareness, and ANC visits. Tolerance values range from 0.584 to 0.978, and VIF values range from 1.023 to 1.711, all within acceptable limits. This indicates no serious multicollinearity, and the regression model remains reliable. Statistical Checks show

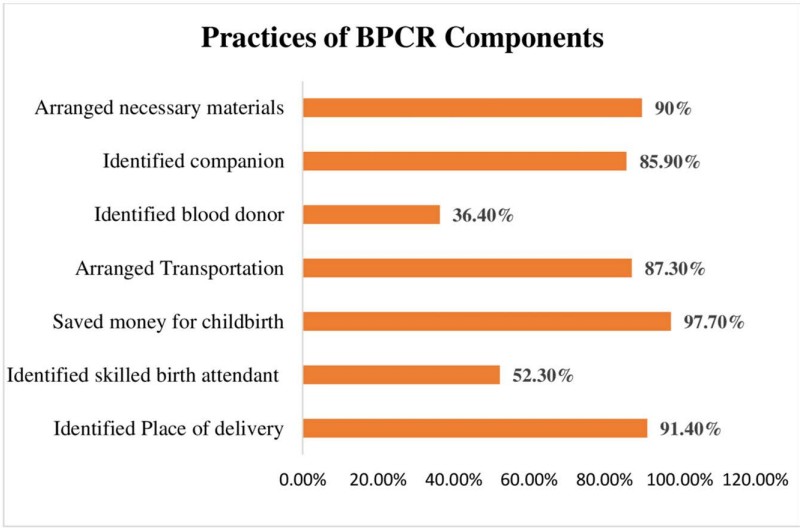

**Fig 2. Birth preparedness practices.** Bar graph illustrating various aspects of childbirth preparedness among women. The orange bars indicate the proportion of women who took specific preparatory actions: 91.4% identified the place of delivery, 97.7% saved money for childbirth, 87.3% arranged transportation, 85.9% selected a companion, and 90% procured necessary materials. In contrast, 52.3% identified skilled birth attendants, and only 36.4% made preparations for blood donors, highlighting a significant gap in blood donor preparedness compared to other aspects of childbirth readiness (Table 5).

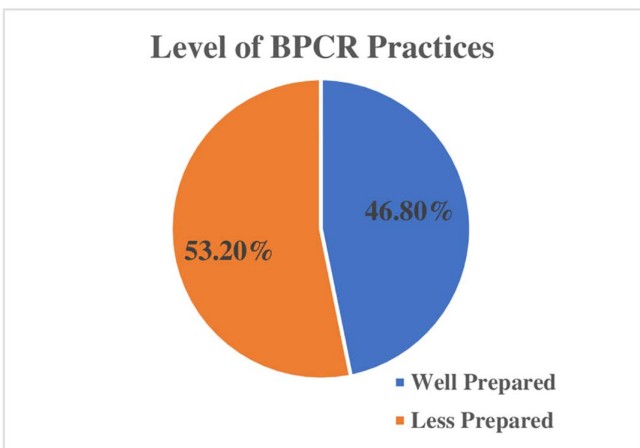

**Fig 3. Level of BPCR practice.** Pie chart depicting the distribution of women's birth preparedness and complication readiness scores. The blue segment represents 103 women (46.8%) who were classified as well-prepared for birth and potential complications, based on key components of birth preparedness and complication readiness. The orange segment represents 117 women (53.2%) who were categorized as less prepared. This distribution highlights the proportion of women with varying levels of preparedness for childbirth and associated complications (Table 7).

no serious multicollinearity, as all variables are within acceptable limits. Though Some knowledge-related variables are moderately correlated, they do not affect the model's reliability.

## Multivariate analysis of factors associated with BPCR practice

Six characteristics that exhibit statistically significant association (p < 0.05) with birth preparedness and complication readiness practice at 95% CI during bivariate analysis were further subjected to multivariate logistic regression for adjustment

**Table 6. Association between level of BPCR practices and explanatory variables among recently delivered woman in Chitwan, Nepal (N = 220).**

| Variables | Level of BPCR practice | | Chi-square | p-value |
|---|---|---|---|---|
| | Well prepared % | Less prepared % | | |
| **Age of Child** | | | | |
| 0-4 | 47 (50) | 47 (50) | 3.596 | 0.166 |
| 5-9 | 21 (36.2) | 37 (63.8) | | |
| 10-12 | 35 (51.5) | 33 (48.5) | | |
| **Mothers age** | | | | |
| 17-25 | 46 (42.2) | 63 (57.8) | 1.849 | 0.174 |
| >25 | 57 (51.4) | 54 (48.6) | | |
| **Family type** | | | | |
| Nuclear | 46 (39.7) | 70(60.3) | 5.057 | **0.025*** |
| Joint/ Extended | 103 (46.8) | 47 (45.2) | | |
| **Ethnicity** | | | | |
| Brahmin/Chhetri | 50 (49.5) | 51 (50.5) | 0.541 | 0.462 |
| Non-Brahmin/Chhetri | 53 (44.5) | 66 (55.5) | | |
| **Religion** | | | | |
| Hindu | 84 (46.4) | 97 (53.6) | 0.069 | 0.793 |
| Non-Hindu | 19 (48.7) | 20 (51.3) | | |
| **Mother education (n = 215)** | | | | |
| General/basic | 27 (43.5) | 35 (56.5) | 0.663 | 0.415 |
| Secondary and above | 76 (49.7) | 77 (50.3) | | |
| **Mother occupation** | | | | |
| Housewife/agriculture | 85 (45) | 104 (55) | 1.833 | 0.176 |
| Other than housewife | 18 (58.1) | 13 (41.9) | | |
| **Monthly income** | | | | |
| Less than 50,000 | 41 (41) | 59 (59) | 2.493 | 0.114 |
| More than 50,000 | 62 (51.7) | 58 (48.3) | | |
| **Age at marriage** | | | | |
| Teenage (≤19) | 36 (45) | 44 (55) | 0.167 | 0.683 |
| Above teenage (>19) | 67 (47.9) | 73 (52.1) | | |
| **Knowledge on components of BPCR** | | | | |
| Adequate | 83 (63.8) | 47 (36.2) | 37.00 | **<0.001*** |
| Inadequate | 20 (22.2) | 70 (77.8) | | |
| **Knowledge on signs and symptoms during pregnancy** | | | | |
| Adequate | 64 (57.1) | 48 (42.9) | 9.768 | **0.002*** |
| Inadequate | 39 (36.1) | 69 (63.9) | | |
| **Knowledge on signs and symptoms during childbirth** | | | | |
| Adequate | 29 (57.1) | 16 (42.9) | 7.059 | **0.008*** |
| Inadequate | 74 (36.1) | 101 (63.9) | | |
| **Knowledge on signs and symptoms during postpartum** | | | | |
| Adequate | 35 (57.4) | 26 (42.6) | 3.779 | **0.05*** |
| Inadequate | 68 (42.8) | 91 (57.2) | | |
| **Knowledge on ANC visit protocol** | | | | |
| Yes | 100 (50.3) | 99 (49.7) | 9.868 | **0.02*** |
| No | 3 (14.3) | 18 (85.7) | | |

*(Continued)*

**Table 6.** (Continued)

| Variables | Level of BPCR practice | | Chi-square | p-value |
|---|---|---|---|---|
| | Well prepared % | Less prepared % | | |
| **Times of child bear** | | | | |
| ≤2 | 91 (45.5) | 109 (54.5) | 1.535 | 0.215 |
| >2 | 12 (60) | 8(40) | | |
| **ANC visit** | | | | |
| Yes | 103 (47.9) | 112 (52.1) | | 0.062[β] |
| No | 0 (0) | 5 (100) | | |
| **Number of ANC visit** | | | | |
| Less than four times | 6 (24) | 19 (76) | 5.535 | **0.019*** |
| Greater than or equal to four times | 93 (48.9) | 97 (51.1) | | |
| **Place of delivery** | | | | |
| Home | 0 (0) | 3 (1.4) | | 0.250 [β] |
| Institutional | 103 (47.5) | 114 (51.8) | | |
| **Complication experiences** | | | | |
| Yes | 48 (47.5) | 53 (52.8) | 0.037 | 0.847 |
| No | 55 (46.2) | 64 (53.8) | | |

*Denotes significant association (By applying Pearson chi-square test at 5% level of significance) and β denotes association by applying Fisher's Exact test at 5% level of significance.

of possible confounder. Multicollinearity test was done. None of them have tolerance <0.1 and VIF > 10. There was no problem of serious collinearity among independent variables.

Birth preparedness practice was found to be statistically significant (p < 0.05) with knowledge on components of BPCR. Women with adequate knowledge on components of BPCR are 5 times more likely to practice or well pre-pared for birth and complication as compared to women with inadequate knowledge (AOR = 5.34,95%CI: (2.728–10.478) (Table 7).

## Discussions

BPCR is one of the proven and effective health care strategy in preventing maternal mortality especially for countries with prevailing high risk of maternal deaths and inefficient health care system. The study found that the overall prevalence of BPCR was 46.8%, which is comparable to rates reported in studies conducted in Southern Nigeria (48.4%) [33] and Southern Ethiopia (48.5%) [10], Sidama (17%), Wolyita (18.3%) zones of Ethiopia [34,35]. However, the prevalence rate was better than that reported in a study conducted in Dang district (44.36%) [21]. On the other hand, the rate was lower than rates reported in studies conducted in Rapti municipality, Kaski district [20,36] and other South Asian countries, including Sri Lanka, Karnataka, India, and West Bengal [37,38], as well as Southwest Nigeria and Tanzania [16,39]. The most common birth preparedness practice observed in this study was saving money, which may be explained by the fact that both women and their partners know that money is required to facilitate referral in case of complications. And it is interesting to note that other studies on birth preparedness have reported similar findings [40,41]. The study found that identification of blood donor as an obstetric emergency was not considered crucial issue by some respondents as other components where only 34% identified potential blood donor. However, other study reported even lower percent of respondents practiced this component [9]. The variation in findings may be attributed to several factors, such as differ-ences in the study population, with our study involving recently delivered women, while the studies in South Ethiopia [34] and Dang [42] included pregnant women and both delivered and pregnant women, respectively. Pregnant women may be

**Table 7. Multivariate analysis of factors associated with BPCR practice among recently delivered woman in Chitwan, Nepal (n = 220).**

| Variable | COR (95% CI) | AOR (95%CI) | p-value |
|---|---|---|---|
| **Family Type** | | | |
| Nuclear | Ref | Ref | 0.88 |
| Joint/Extended | 1.846(1.080-3.155) | 1.689(0.926-3.080) | |
| **Knowledge on components of BPCR** | | | |
| Inadequate | Ref | Ref | **0.00** |
| Adequate | 6.181(3.351-11.402) | **5.347(2.728-10.478) *** | |
| **Knowledge on key danger signs during pregnancy** | | | |
| Inadequate | Ref | Ref | |
| Adequate | 2.359(1.371-4.058) | 1.332(0.669-2.652) | 0.414 |
| **Knowledge on key danger signs during labor** | | | |
| Inadequate | Ref | Ref | 0.675 |
| Adequate | 2.474(1.253-4.883) | 1.213(0.492-2.991) | |
| **Knowledge on key danger signs during postpartum** | | | |
| Inadequate | Ref | Ref | 0.605 |
| Adequate | 1.801(0.992-3.272) | 0.801(0.346-1.856) | |
| **Knowledge on ANC visit as** | | | |
| Less than four times | Ref | Ref | 0.14 |
| Greater than four times | 2.499(1.451-4.305) | 2.151(1.170-3.954) | |

*Variables significant at p<0.05 *Ref*: Reference Category CI: Confidence Interval OR: Odds Ratio.

unable to report their preparedness for situations they have not yet experienced. Additionally, variations in female literacy levels, empowerment, study setting, socio-cultural difference, and methodological differences in assessing BPCR could also contribute to the differences in findings.

Nearly three-quarters (72.1%) and over one-third (45.3%) of the study were knowledgeable about identifying place of delivery and arranging transportation as essential components of BPCR, this aligns with similar findings reported in other studies. In a survey conducted in Samara Logia Town, where 75% and 45% of participants were knowledgeable on those components [43]. The study revealed that 59.1%, 50.9%, of participants had knowledge about the components of BPCR and danger signs during pregnancy This finding is in contrast to a study conducted in Farta district, Ethiopia, which reported a significantly lower percentage of 14.3% and 23.3% of participants with knowledge of the components and danger signs during pregnancy respectively [11]. The study revealed that the participants' understanding of danger signs during labor/childbirth and the postpartum period was inadequate, with only 20.5% and 27.3% of them reporting familiarity with these signs, respectively which is consistent with the study done in Sodo town, Southern Ethiopia [10]. The finding that severe bleeding was identified as a major sign and symptom during childbirth is similar in both the current study and a study conducted in Mizan-Aman, Southwest Ethiopia [39]. Women having knowledge of obstetric danger signs might have chance to improve their decision-making ability on their maternal and child health care service utilization and can respond immediately in the event of emergencies to avoid delays [11].

Approximately Ninety Eight Percent of the women have had at least one ANC visit, about half (47.3%) attended the minimum recommended number of four or more visits, was found to be low as compared to the national NDHS data (81%) [44] and studies in Kassena- Nankana of Ghana [45]. This study included a specific decision-making variable which is absent from most studies. But in this study almost half of them made decision jointly regarding the place of delivery as in the study in Rapti municipality [32].

Our study identified that the level of BPCR practice is significantly associated with family type ($\chi^2 = 5.057$, p = 0.025)and knowledge on ANC visit as protocol ($\chi^2 = 9.868$, p = 0.02) which is compliance with the findings done in Kenya and Tanzania and Rapti Municipality of Chitwan, Nepal [32,46]. Similarly, knowledge on components of BPCR (($\chi^2 = 37$, p = 0.001), danger sign and symptoms during pregnancy ($\chi^2 = 9.768$, p = 0.002), labor/childbirth and postpartum ($\chi^2 = 3.779$, p = 0.05) are significantly associated with BPCR practice, which is similar with the findings of community- based study done in rural area in Bangladesh, Mizan-Aman Town, Southwest Ethiopia and Nepal where association between knowledge and practice was observed [6,22]. Also, the current study found that the BPCR practice was statistically significant with number of ANC visits. This finding is supported by the study in central Tanzania where woman who had attended 4 or more times antenatal care were better prepared than women who had attended <4 times antenatal care [47]. The significant influence of woman's literacy level on BPCR practice is highlighted by many studies from Sri-lanka and African countries [48]. However, in our study and a study in Biratnagar, Nepal, Karnataka India, Bangladesh, woman's education level did not show a significant association with BPCR [15,40,49]. This could be attributed to overall high (97.7%) literacy rate in this study population.

Having the knowledge of BPCR increases the practice of BPCR among recently delivered women and 5 times more likely to be well prepared according to this study. This is comparable with the studies from Central Ethiopia [50] Bench Maji zone [51] and Farta district [11]. The inference of this finding could be once women become knowledgeable about BPCR, they are expected to practice the components of BPCR and ready to act upon it when it occurs.

The advantage of asking women who are recently delivered about the birth preparedness as they have completed their pregnancies and already made all necessary arrangement. As there may be variation in their service use in pregnant women, they will not provide the information based on their plan to use those services.

## Conclusions

Although not satisfactory in the view of expectations, a relatively higher practice of BPCR had been observed in the study area compared to other reports. Majority of participants saved money as one of preparation for childbirth while only few prepared blood donors in case of obstetric complication and emergency. This study also revealed that the respondent's knowledge of danger sign and symptoms during pregnancy was adequate but during child birth and postpartum was low. Similarly, the women having recommended ANC visit was low as compared to national data. These findings lend support to reinforce health education on BPCR during pregnancy. Further the principal factors affecting the BPCR practice were family type, ANC visit, knowledge on components of BPCR, danger signs during pregnancy, labor/Childbirth, and postpartum period, which suggests more intervention to be taken at health institution and health care providers. Furthermore, pregnant women should continuously be encouraged to attend ANC as per the protocol so that interventions such as Sensitization of pregnant women, during ANC visit by the health workers, regarding Components of BPCR, danger sign and symptoms during pregnancy, may enhance BPCR.

**Implication of the study**

**Strengthen health education during ANC visits**, as knowledge of BPCR and danger signs significantly improves practice.

**Promote full ANC attendance (4 or more visits)** to enhance maternal preparedness and reduce delays in seeking care.

**Develop targeted interventions** for low-practiced BPCR components, such as identifying a potential blood donor.

**Involve partners and family members** in maternal health education to support joint decision-making on place of delivery and emergency planning.

## Study limitations

• Self -reported practice may compromise the validity of the findings.

• There could also be information bias as the questions were asked to all those who delivered in the last 12 months. Some might not have recalled the danger signs, and symptoms.

## Supporting information

**S1 File. Final data 101.**
(SAV)

## Acknowledgments

We would like to thank school of public health, Chitwan Medical college, health section of Bharatpur metropolitan city and health institutions for providing support in undertaking this research.

## Author contributions

**Conceptualization:** Asmita Ghimire, Bipan Bahadur Tiwari, Eak Narayan Poudel, Laxman Poudel.

**Data curation:** Asmita Ghimire, Bipan Bahadur Tiwari, Laxman Poudel.

**Formal analysis:** Asmita Ghimire, Bipan Bahadur Tiwari.

**Funding acquisition:** Asmita Ghimire, Devraj Regmi.

**Investigation:** Asmita Ghimire, Bipan Bahadur Tiwari.

**Methodology:** Asmita Ghimire, Bipan Bahadur Tiwari, Eak Narayan Poudel, Mamta Chhetri, Devraj Regmi, Laxman Poudel.

**Resources:** Asmita Ghimire, Bipan Bahadur Tiwari, Devraj Regmi.

**Software:** Asmita Ghimire, Bipan Bahadur Tiwari.

**Supervision:** Bipan Bahadur Tiwari, Eak Narayan Poudel.

**Validation:** Eak Narayan Poudel, Mamta Chhetri, Devraj Regmi.

**Visualization:** Bipan Bahadur Tiwari.

**Writing – original draft:** Asmita Ghimire, Eak Narayan Poudel.

**Writing – review & editing:** Bipan Bahadur Tiwari.

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
