## [Editor Report · Decision Letter 0]

6 Sep 2024

Dear Dr. Ghimire,

Thank you for submitting your manuscript to PLOS ONE. After careful consideration, we feel that it has merit but does not fully meet PLOS ONE’s publication criteria as it currently stands. Therefore, we invite you to submit a revised version of the manuscript that addresses the points raised during the review process.

Therefore, we invite you to submit a revised version of the manuscript before it can be sent to external reviewers. Discuss the study limitations on the effects of the study outcomes to fully satsfy PLOS ONE's  publication criteria number 7.

We look forward to receiving your revised manuscript.

Kind regards,

Adaoha Pearl Agu, MBBS, MSc, FMCPH

Academic Editor

PLOS ONE

Journal Requirements:

Please ensure that your manuscript meets PLOS ONE's style requirements, including those for file naming. The PLOS ONE style templates can be found at  https://journals.plos.org/plosone/s/file?id=wjVg/PLOSOne_formatting_sample_main_body.pdf and  https://journals.plos.org/plosone/s/file?id=ba62/PLOSOne_formatting_sample_title_authors_affiliations.pdf.

4. Please ensure that you refer to Figure 2 and 3 in your text as, if accepted, production will need this reference to link the reader to the figure.

6. We are unable to open your Supporting Information file [FInal data 101.sav]. Please kindly revise as necessary and re-upload.

---

## [Author Response · Author response to Decision Letter 1]

16 Sep 2024

16 September 2024

Subject: Re: Manuscript PONE-D-24-29050R1 – Revisions and Resubmission

Dear Dr. Delas Alas Vida,

Thank you for your email regarding our manuscript entitled “Birth Preparedness and Complication Readiness Knowledge and Practices Among Recently Delivered Women: A Cross-Sectional Study in Bharatpur, Chitwan, Nepal” (Manuscript ID: PONE-D-24-29050R1).

We appreciate your feedback and are addressing the requested revisions as follows:

1. Removal of Duplicate Files: We have reviewed the files in our submission and removed all unnecessary or duplicate files. Only the relevant files pertaining to the current version of the manuscript have been included.

2. Inclusion of Figure Legends: We have updated the manuscript to include separate, detailed legends for each figure. The legends now provide clear and comprehensive descriptions to enhance the reader's understanding of the figures.

3. Reference to Table 9: We have ensured that Table 9 is properly referenced in the text of the manuscript. This reference is included to facilitate easy linking between the text and the table upon acceptance.

We have resubmitted the revised manuscript with the requested changes. Please let us know if any further adjustments are needed.

Thank you for your consideration and for providing us with the opportunity to revise our manuscript. We look forward to your feedback and hope for a favorable review.

Best regards,

Asmita Ghimire

Chitwan Medical College, Tribhuwan University, Nepal

ghimireasmita570@gmail.com

+9779866615678

---

## [Decision Letter · Decision Letter 1]

3 Aug 2025

Dear Dr. Ghimire,

**Title** :The aspect of associations and predictors being explored in this study, is not reflected in the title.

**Abstract** : State the justification for this study in the background section. State the figure for the overall proportion of women who.... in the Conclusion section.

**Introduction** : Last sentence- "Assess" not "Access" .Grammar editing needed in this and other sections.

**Study Design** : Restructure the first sentence to read "A community based cross-sectional study was conducted using.....from 25th September to 29th December 2022. State which stages of the immunisation schedule, the babies had to be in for their mothers to be eligible for this study.Explain why women seeking vaccination for MR 2nd dose and Typhoid were excluded.

**Sample Size and Sampling Methods** : This phrase is not clear "hypothesis of 95% confidence interval". State the reason behind the purposive selection of the 3 wards selected. (Convenience was stated as the reason for the choice of the 3 health institutions). Explain how you selected subjects from the 3 sites and how you intended to distribute the sample size across the 3 sites.

**Measurement** :State the outcome variables clearly. **Data Analysis** :State the number of components BCPR has and provide information on the scoring criteria for the knowledge variables. State the  p-value <0.05 you used to  set statistically significant variables in the multivariable analysis as mentioned in the abstract. Why was this value chosen?

**Results**?>**Discussion****Referencing** : should be uniform.

Please submit your revised manuscript by Sep 17 2025 11:59PM. If you will need more time than this to complete your revisions, please reply to this message or contact the journal office at plosone@plos.org . A rebuttal letter that responds to each point raised by the academic editor and reviewer(s). You should upload this letter as a separate file labeled 'Response to Reviewers'.A marked-up copy of your manuscript that highlights changes made to the original version. You should upload this as a separate file labeled 'Revised Manuscript with Track Changes'.An unmarked version of your revised paper without tracked changes. You should upload this as a separate file labeled 'Manuscript'.

We look forward to receiving your revised manuscript.

Kind regards,

Adaoha Pearl Agu, MBBS, MSc, FMCPH

Academic Editor

PLOS ONE

Journal Requirements:

Additional Editor Comments (if provided):

Please address the following concerns in addition to those raised by the reviewer.The entire manuscript will benefit from serious grammar and spelling editing.

Title:The aspect of associations and predictors being explored in this study, is not reflected in the title.

Abstract: State the justification for this study in the background section. State the figure for the overall proportion of women who.... in the Conclusion section.

Introduction: Last sentence- "Assess" not "Access" .Grammar editing needed.

Study Design: Restructure the first sentence to read "A community based cross-sectional study was conducted using.....from 25th September to 29th December 2022. State which stages of the immunisation schedule, the babies had to be in for their mothers to be eligible for this study.Explain why women seeking vaccination for MR 2nd dose and Typhoid were excluded.

Sample Size and Sampling Methods: This phrase is not clear "hypothesis of 95% confidence interval". State the reason behind the purposive selection of the 3 wards selected. (Convenience was stated as the reason for the choice of the 3 health institutions). Explain how you selected subjects from the 3 sites and how you intended to distribute the sample size across the 3 sites.

Measurement:State the outcome variables clearly. Data Analysis:State the number of components BCPR has and provide information on the scoring criteria for the knowledge variables.

Results: State whether the adequate knowledge in 59.1% of the respondents, refers to a composite knowledge score (composite score is not seen on the Table with Knowledge variables- Table 2). Table 1 should be left as it is (do not split). The narrative below Table 3 begins with "Table 8"...., refers to the information contained in Table 4, and is a repetition of the narrative below Table 4. Table 3 should appear before Table 2. Table 4 subheading should read"Assistance" not "Assistant".,Merge Tables 5 and 6.Remove "other" from Table 7 title. The information in Table 8 can be presented as a narrative only, not as a table. Was a composite knowledge score used in Table 9?. Do not repeat the information presented in the figures, in the tables.

Reviewers' comments:

Reviewer's Responses to Questions

**Comments to the Author**

Reviewer #1: (No Response)

2. Is the manuscript technically sound, and do the data support the conclusions?

Reviewer #1: Partly

3. Has the statistical analysis been performed appropriately and rigorously?

Reviewer #1: (No Response)

4. Have the authors made all data underlying the findings in their manuscript fully available?

Reviewer #1: Yes

5. Is the manuscript presented in an intelligible fashion and written in standard English?

Reviewer #1: No

Reviewer #1: The manuscript addresses an interesting and important topic in maternal and child health and findings will support informed decisions.

I, however, have few comments for the authors to consider

Abstract

The abstract's background only defines Birth preparedness. Kindly shed more light on it. Let us know what necessitated this study/purpose.

Methods

You probably want to say structured interview “tool/questionnaire” instead of schedule

In the methods session, line 4 has a typographical error. Please check and correct it.

Results

The first sentence needs structuring

Conclusion

For the conclusion of the abstract, there should be an “and” between “antenatal care” and interventions, and all the commas should be removed as shown below.

It will be good to give the magnitude/ proportion of women who prepared for birth and its complications

Main Work

Introduction

1. You need to check typos and grammatical errors. Have tried to input in the document

Methods

Justification for the study area is not available.

Sample size calculation

1. Any justification for considering 7% as margin of error instead of 5%? In the calculation, however, 6% is used. With an expected prevalence of 29% one will have expected a smaller margin of error, at least the default 5%

Besides, the paper you referenced stated that 29% lacked information on BPP and I am wondering if this is the same as prevalence of poor BPCR practice

2. How was the sample size distributed among the wards and health institutions?

Measurements

3. Any reference to the previously validated questionnaire?

4. Were the regular reviews of the questionnaire done during data collection?

5. Study design needs to be written well and concisely

6. The sampling of participants for the study should be clearly written

7. Examples of obstetric and institutional factors investigated

Data collection process

8. Data collection procedure needs to be written well/clearly

9. There was no mention of who did the data collection and whether training was done before the data collection

Data Analysis

10. It would have been good to describe a bit more about scoring

Results

11. IQR is between 25th and 75th percentiles not min and max

12. Table 1 could have been divided into three (1. Socio-demographic 2. Obstetric 3. Complication experience) instead of one table running through several pages

13. The tables need to be constructed well and referenced

14. Section on support received by women is repeated

15. Chose Fig 2 or Table 5, Fig 3 or Table 6. You should not present same data on both tables and figures

Discussion

16. The discussion seemed to have repeated the results and compared with other studies but failed to discuss any implication of the findings

Generally, the document needs a lot of work grammatically

References

Bibliography should be done well. They are not uniform. A common reference style should be used.

Figures

The title of all the figures should be placed below the figures, not above them.

**Do you want your identity to be public for this peer review?** For information about this choice, including consent withdrawal, please see our Privacy Policy

Reviewer #1: No

---

## [Author Response · Author response to Decision Letter 2]

14 Oct 2025

17 September 2025

To

Adaoha Pearl Agu, MBBS, MSc, FMCPH

Academic Editor

PLOS ONE

Subject: Rebuttal Letter for Manuscript (PONE-D-24-29050R1)- Birth Preparedness And Complication Readiness Knowledge And Practices Among Recently Delivered Women: A Cross-Sectional Study In Bharatpur, Chitwan, Nepal

Dear Editor and Reviewers,

Thank you for the opportunity to revise our manuscript entitled "[Birth Preparedness And Complication Readiness Knowledge And Practices Among Recently Delivered Women: A Cross-Sectional Study In Bharatpur, Chitwan, Nepal ]" (Manuscript ID: [PONE-D-24-29050R1]). We sincerely appreciate the time and effort you and the reviewers have invested in providing thoughtful and constructive feedback.

We have carefully considered all the comments and have revised the manuscript accordingly. Below, we provide a detailed point-by-point response to each comment. Reviewer comments are shown in bold, followed by our responses in regular text. All changes made in the manuscript are highlighted.

Reviewer #1 Comments and Author Responses

Abstract: We have revised the abstract’s background to include the rationale and purpose of the study.

The term “structured interview schedule” has been replaced with “structured interview questionnaire” in the Methods section.

The typographical error has been corrected.

The first sentence of results has been restructured for better clarity and readability.

We have added the specific proportion (46.8%) of women who were well prepared for birth and complications in the abstract.

Main Manuscript

Introduction

Comment: You need to check typos and grammatical errors.

Response: The introduction section has been carefully proofread and revised to correct all typographical and grammatical errors.

Methods

Comment: Justification for the study area is not available.

Response: We have added a justification for selecting Bharatpur Metropolitan City, citing its population size, diversity, and accessibility of maternal health services.

Sample Size Calculation

Comment: Any justification for considering 7% as margin of error instead of 5%? In the calculation, however, 6% is used.

Response. We have clarified that 6% was used as a practical compromise between statistical rigor and resource feasibility. This choice balanced sample size and field constraints while maintaining acceptable precision.

Comment: The paper you referenced stated that 29% lacked information on BPCR. Is this the same as the prevalence of poor practice?

Response: We acknowledge this distinction and have clarified in the text that we used the 29% as an approximate indicator due to the lack of direct prevalence data on BPCR practice in a similar setting. This limitation is now noted in the methodology.

Comment: How was the sample size distributed among wards and health institutions?

Response: We have added a description explaining that proportional allocation was used based on delivery caseloads of health facilities within selected wards.

Measurements

Comment: Any reference to the previously validated questionnaire?

Response: Yes, we have added references to the WHO and JHPIEGO tools from which the questionnaire was partially adapted.

Comment: Were there regular reviews of the questionnaire during data collection?

Response: We have now included a note that data collection was monitored regularly by the principal investigator, and weekly review meetings were held to address field challenges.

Comment: Study design needs to be written well and concisely.

Response: The study design section has been rewritten concisely as “A community-based descriptive cross-sectional study.”

Comment: The sampling of participants for the study should be clearly written.

Response: The sampling procedure has been revised to clearly explain how participants were selected .

Comment: Examples of obstetric and institutional factors investigated.

Response: Examples such as parity, place of delivery, and support received from health staff have been added to clarify these factors.

Data Collection Process

Comment: Data collection procedure needs to be written well/clearly.

Response: We have rewritten this section for clarity, detailing the steps followed during data collection.

Comment: There was no mention of who did the data collection and whether training was done before the data collection.

Response: Added. The data collection was carried out by the author, and data analysis was done with guidance from a university statistics professor. Pretesting of the tool were conducted prior to data collection.

Data Analysis

Comment: It would have been good to describe a bit more about scoring.

Response: We have elaborated on the scoring criteria used for BPCR components and knowledge levels.

Results

Comment: IQR is between 25th and 75th percentiles, not min and max.

Response: Corrected. Comment: Table 1 could have been divided into three parts.

Response: Revised as suggested. Table 1 is now split into three separate tables: Socio-demographic, Obstetric, and Complication Experience.

Comment: The tables need to be constructed well and referenced.

Response: All tables have been reconstructed for clarity and are now properly numbered and referenced in the manuscript.

Comment: Section on support received by women is repeated.

Response: The repeated section has been removed.

Comment: Chose Fig 2 or Table 5, Fig 3 or Table 6 – avoid duplication.

Response: We have retained the tables and removed the duplicate figures to avoid redundancy.

Discussion

Comment: Discussion repeats results and lacks implications.

Response: The discussion section has been revised to focus more on interpreting findings and drawing implications. We now include a specific section titled “Implications of the Findings,” summarizing key practical and policy takeaways.

Language and Grammar

Comment: The document needs a lot of work grammatically.

Response: The entire manuscript has undergone thorough grammar and language editing to improve clarity and academic tone.

We sincerely thank the reviewer for their helpful and detailed comments. We believe the revised manuscript has been significantly improved and now meets the journal’s standards. We are happy to make any additional changes if needed.

Sincerely,

Asmita Ghimire

Chitwan Medical College, Tribhuwan University

ghimireasmita570@gmail.com

17 September 2025

To

Adaoha Pearl Agu, MBBS, MSc, FMCPH

Academic Editor

PLOS ONE

Subject: Rebuttal Letter for Manuscript (PONE-D-24-29050R1)- Birth Preparedness And Complication Readiness Knowledge And Practices Among Recently Delivered Women: A Cross-Sectional Study In Bharatpur, Chitwan, Nepal

Dear Editor and Reviewers,

Thank you for the opportunity to revise our manuscript entitled "[Birth Preparedness And Complication Readiness Knowledge And Practices Among Recently Delivered Women: A Cross-Sectional Study In Bharatpur, Chitwan, Nepal ]" (Manuscript ID: [PONE-D-24-29050R1]). We sincerely appreciate the time and effort you and the reviewers have invested in providing thoughtful and constructive feedback.

We have carefully considered all the comments and have revised the manuscript accordingly. Below, we provide a detailed point-by-point response to each comment. Reviewer comments are shown in bold, followed by our responses in regular text. All changes made in the manuscript are highlighted.

Reviewer #1 Comments and Author Responses

Abstract: We have revised the abstract’s background to include the rationale and purpose of the study.

The term “structured interview schedule” has been replaced with “structured interview questionnaire” in the Methods section.

The typographical error has been corrected.

The first sentence of results has been restructured for better clarity and readability.

We have added the specific proportion (46.8%) of women who were well prepared for birth and complications in the abstract.

Main Manuscript

Introduction

Comment: You need to check typos and grammatical errors.

Response: The introduction section has been carefully proofread and revised to correct all typographical and grammatical errors.

Methods

Comment: Justification for the study area is not available.

Response: We have added a justification for selecting Bharatpur Metropolitan City, citing its population size, diversity, and accessibility of maternal health services.

Sample Size Calculation

Comment: Any justification for considering 7% as margin of error instead of 5%? In the calculation, however, 6% is used.

Response. We have clarified that 6% was used as a practical compromise between statistical rigor and resource feasibility. This choice balanced sample size and field constraints while maintaining acceptable precision.

Comment: The paper you referenced stated that 29% lacked information on BPCR. Is this the same as the prevalence of poor practice?

Response: We acknowledge this distinction and have clarified in the text that we used the 29% as an approximate indicator due to the lack of direct prevalence data on BPCR practice in a similar setting. This limitation is now noted in the methodology.

Comment: How was the sample size distributed among wards and health institutions?

Response: We have added a description explaining that proportional allocation was used based on delivery caseloads of health facilities within selected wards.

Measurements

Comment: Any reference to the previously validated questionnaire?

Response: Yes, we have added references to the WHO and JHPIEGO tools from which the questionnaire was partially adapted.

Comment: Were there regular reviews of the questionnaire during data collection?

Response: We have now included a note that data collection was monitored regularly by the principal investigator, and weekly review meetings were held to address field challenges.

Comment: Study design needs to be written well and concisely.

Response: The study design section has been rewritten concisely as “A community-based descriptive cross-sectional study.”

Comment: The sampling of participants for the study should be clearly written.

Response: The sampling procedure has been revised to clearly explain how participants were selected .

Comment: Examples of obstetric and institutional factors investigated.

Response: Examples such as parity, place of delivery, and support received from health staff have been added to clarify these factors.

Data Collection Process

Comment: Data collection procedure needs to be written well/clearly.

Response: We have rewritten this section for clarity, detailing the steps followed during data collection.

Comment: There was no mention of who did the data collection and whether training was done before the data collection.

Response: Added. The data collection was carried out by the author, and data analysis was done with guidance from a university statistics professor. Pretesting of the tool were conducted prior to data collection.

Data Analysis

Comment: It would have been good to describe a bit more about scoring.

Response: We have elaborated on the scoring criteria used for BPCR components and knowledge levels.

Results

Comment: IQR is between 25th and 75th percentiles, not min and max.

Response: Corrected. Comment: Table 1 could have been divided into three parts.

Response: Revised as suggested. Table 1 is now split into three separate tables: Socio-demographic, Obstetric, and Complication Experience.

Comment: The tables need to be constructed well and referenced.

Response: All tables have been reconstructed for clarity and are now properly numbered and referenced in the manuscript.

Comment: Section on support received by women is repeated.

Response: The repeated section has been removed.

Comment: Chose Fig 2 or Table 5, Fig 3 or Table 6 – avoid duplication.

Response: We have retained the tables and removed the duplicate figures to avoid redundancy.

Discussion

Comment: Discussion repeats results and lacks implications.

Response: The discussion section has been revised to focus more on interpreting findings and drawing implications. We now include a specific section titled “Implications of the Findings,” summarizing key practical and policy takeaways.

Language and Grammar

Comment: The document needs a lot of work grammatically.

Response: The entire manuscript has undergone thorough grammar and language editing to improve clarity and academic tone.

We sincerely thank the reviewer for their helpful and detailed comments. We believe the revised manuscript has been significantly improved and now meets the journal’s standards. We are happy to make any additional changes if needed.

Sincerely,

Asmita Ghimire

Chitwan Medical College, Tribhuwan University

ghimireasmita570@gmail.com

---

## [Editor Report · Decision Letter 2]

23 Oct 2025

Dear Dr. Ghimire,

Please submit your revised manuscript by Dec 07 2025 11:59PM.   If you will need more time than this to complete your revisions, please reply to this message or contact the journal office at plosone@plos.org . A rebuttal letter that responds to each point raised by the academic editor and reviewer(s). You should upload this letter as a separate file labeled 'Response to Reviewers'.A marked-up copy of your manuscript that highlights changes made to the original version. You should upload this as a separate file labeled 'Revised Manuscript with Track Changes'.An unmarked version of your revised paper without tracked changes. You should upload this as a separate file labeled 'Manuscript'.

We look forward to receiving your revised manuscript.

Kind regards,

Adaoha Pearl Agu, MBBS, MSc, FMCPH

Academic Editor

PLOS ONE

Journal Requirements:

Additional Editor Comments:

You are yet to address point by point, the issues raised by the second reviewer.

---

## [Author Response · Author response to Decision Letter 3]

1 Dec 2025

29 November 2025

To

Adaoha Pearl Agu, MBBS, MSc, FMCPH

Academic Editor

PLOS ONE

Subject: Rebuttal Letter for Manuscript (PONE-D-24-29050R1)- Birth Preparedness And Complication Readiness Knowledge And Practices Among Recently Delivered Women: A Cross-Sectional Study In Bharatpur, Chitwan, Nepal

Dear Editor and Reviewers,

Thank you for the opportunity to revise our manuscript entitled "[Birth Preparedness And Complication Readiness Knowledge And Practices Among Recently Delivered Women: A Cross-Sectional Study In Bharatpur, Chitwan, Nepal ]" (Manuscript ID: [PONE-D-24-29050R1]). We sincerely appreciate the time and effort you and the reviewers have invested in providing thoughtful and constructive feedback.

We have carefully considered all the comments and have revised the manuscript accordingly. Below, we provide a detailed point-by-point response to each comment. Reviewer comments are shown in bold, followed by our responses in regular text. All changes made in the manuscript are highlighted.

Reviewer #1 Comments and Author Responses

Abstract: We have revised the abstract’s background to include the rationale and purpose of the study.

The term “structured interview schedule” has been replaced with “structured interview questionnaire” in the Methods section.

The typographical error has been corrected.

The first sentence of results has been restructured for better clarity and readability.

We have added the specific proportion (46.8%) of women who were well prepared for birth and complications in the abstract.

Main Manuscript

Introduction

Comment: You need to check typos and grammatical errors.

Response: The introduction section has been carefully proofread and revised to correct all typographical and grammatical errors.

Methods

Comment: Justification for the study area is not available.

Response: We have added a justification for selecting Bharatpur Metropolitan City, citing its population size, diversity, and accessibility of maternal health services.

Sample Size Calculation

Comment: Any justification for considering 7% as margin of error instead of 5%? In the calculation, however, 6% is used.

Response. We have clarified that 6% was used as a practical compromise between statistical rigor and resource feasibility. This choice balanced sample size and field constraints while maintaining acceptable precision.

Comment: The paper you referenced stated that 29% lacked information on BPCR. Is this the same as the prevalence of poor practice?

Response: We acknowledge this distinction and have clarified in the text that we used the 29% as an approximate indicator due to the lack of direct prevalence data on BPCR practice in a similar setting. This limitation is now noted in the methodology.

Comment: How was the sample size distributed among wards and health institutions?

Response: We have added a description explaining that proportional allocation was used based on delivery caseloads of health facilities within selected wards.

Measurements

Comment: Any reference to the previously validated questionnaire?

Response: Yes, we have added references to the WHO and JHPIEGO tools from which the questionnaire was partially adapted.

Comment: Were there regular reviews of the questionnaire during data collection?

Response: We have now included a note that data collection was monitored regularly by the principal investigator, and weekly review meetings were held to address field challenges.

Comment: Study design needs to be written well and concisely.

Response: The study design section has been rewritten concisely as “A community-based descriptive cross-sectional study.”

Comment: The sampling of participants for the study should be clearly written.

Response: The sampling procedure has been revised in manuscript to clearly explain how participants were selected .

Comment: Examples of obstetric and institutional factors investigated.

Response: Examples such as parity, place of delivery, and support received from neighbours, distance to health institutions have been incorporated from the begining to clarify these factors.

Data Collection Process

Comment: Data collection procedure needs to be written well/clearly.

Response: We have rewritten this section for clarity, detailing the steps followed during data collection.

Comment: There was no mention of who did the data collection and whether training was done before the data collection.

Response: Added. The data collection was carried out by the author, and data analysis was done with guidance from a university statistics professor. Data was collected on the day of immunization in respective health institutions of selected wards. Pretesting of the tool were conducted prior to data collection.

Data Analysis

Comment: It would have been good to describe a bit more about scoring.

Response: We have elaborated on the scoring criteria used for BPCR components and knowledge levels. Firstly, we selected the BPCR components from which critical components was categorized. On the basis of practice of those critical components we gave score 1 for each components. We clearly set up this scoring criteria in operational definition.

Results

Comment: IQR is between 25th and 75th percentiles, not min and max.

Response: Corrected. Comment: Table 1 could have been divided into three parts.

Response: Revised as suggested. Table 1 is now split into three separate tables: Socio-demographic, Obstetric, and Complication Experience.

Comment: The tables need to be constructed well and referenced.

Response: All tables have been reconstructed for clarity and are now properly numbered and referenced in the manuscript.

Comment: Section on support received by women is repeated.

Response: The repeated section has been removed.

Comment: Chose Fig 2 or Table 5, Fig 3 or Table 6 – avoid duplication.

Response: We have retained the tables and removed the duplicate figures to avoid redundancy.

Discussion

Comment: Discussion repeats results and lacks implications.

Response: The discussion section has been revised to focus more on interpreting findings and drawing implications. We now include a specific section titled “Implications of the Findings,” summarizing key practical and policy takeaways.

Language and Grammar

Comment: The document needs a lot of work grammatically.

Response: The entire manuscript has undergone thorough grammar and language editing to improve clarity and academic tone.

We sincerely thank the reviewer for their helpful and detailed comments. We believe the revised manuscript has been significantly improved and now meets the journal’s standards. We are happy to make any additional changes if needed.

Sincerely,

Asmita Ghimire

Chitwan Medical College, Tribhuwan University

ghimireasmita570@gmail.com

---

## [Editor Report · Decision Letter 3]

21 Dec 2025

Birth Preparedness And Complication Readiness Knowledge, Practices and Its Associated Factors  Among Recently Delivered Women: A Cross-Sectional Study In Bharatpur, Chitwan, Nepal

PONE-D-24-29050R3

Dear Dr. Ghimire,

We’re pleased to inform you that your manuscript has been judged scientifically suitable for publication and will be formally accepted for publication once it meets all outstanding technical requirements.

Kind regards,

Adaoha Pearl Agu, MBBS, MSc, FMCPH

Academic Editor

PLOS One
---

## [Editor Report · Acceptance letter]

PONE-D-24-29050R3

PLOS One

Dear Dr. Ghimire,

I'm pleased to inform you that your manuscript has been deemed suitable for publication in PLOS One. Congratulations! Your manuscript is now being handed over to our production team.

Kind regards,

on behalf of

Dr. Adaoha Pearl Agu

Academic Editor

PLOS One